# Stakeholder Discourse on Synthetic Fuels: A Positioning and Narrative Analysis

**Dirk Scheer *** and **Lisa Schmieder**

Institute for Technology Assessment and Systems Analysis (ITAS), Karlsruhe Institute for Technologies (KIT), Karlstraße 11, 76133 Karlsruhe, Germany
* Correspondence: dirk.scheer@kit.edu

**Abstract:** The transition of mobility (in German "Verkehrswende") as a fundamental part of the overall energy transition is a controversial field among stakeholders—in particular when it comes to synthetic fuels. There are considerable opposing views on the pros and cons of synthetic fuels within stakeholder communication. Against this background, the aim of this study was to research stakeholder positions and communication by identifying, systemizing, and assessing the bandwidth of stakeholder statements and views in Germany using a document-based positioning analysis. The objective was to provide the broadest possible range of (controversial) assessments on synthetic fuels' future pathways. Based on a document analysis of 41 sources published by 17 stakeholders from the areas of economy, environment, and civil society in the last ten years, we analyzed commonalities and differences in the assessments of the synthetic fuels' path as well as the reasons behind it. The results were synthesized in three narrative frames dominating the German discourse on synthetic fuels, namely: (1) synthetic fuels as a key component for the mobility transition; (2) synthetic fuels as an essential strategic niche management component, and (3) mobility transition as sustainable, affordable, safe, and comfortable mobility—with or without synthetic fuels.

**Keywords:** renewable fuels (refuels); synthetic fuels; stakeholder positioning; discourse analysis

## 1. Introduction

The transformation of the energy and transport system towards decarbonization and defossilization, in order to protect the climate, is a task for societies as a whole, to which politics, business, and society have to contribute for its success. However, societal debate and decision-making reveal often differing viewpoints, interests, and strategies. As such, very different assessments among the stakeholders about the energy transition and its promising transformation pathway can be observed. This is also the case in the field of synthetic fuels. Synthetic fuels stand for renewable fuels and can be defined at a broad scale summarizing all types of fuels (e.g., diesel, gasoline, kerosene) produced on the basis of regenerative sources (i.e., energy and biomass).

From a social science perspective, it is essential to research in advance existing attitudes and positioning of stakeholders and find out which transformation pathways and measures meet approval or rejection among stakeholders from business and industry, environment, and civil society. In the scientific and public debate, several terms of what can be labeled "new fuels" appear such as alternative fuels, synthetic fuels, biofuels, or e-fuels. In our understanding, alternative fuels serve as an umbrella term for a group of fuels that (partly) substitute fossil oil sources for combustion engines. The most prominent alternative fuels are biofuels, which are any fuel derived from biomass, and synthetic fuels or e-fuels that are produced using carbon dioxide (or carbon monoxide) together with hydrogen (at best produced as green hydrogen from renewable power resources).

The analysis of stakeholder communication on new emerging technologies is an important field in social-science-based risk and technology research [1,2]. It relates to

technology acceptance and acceptability research with a focus on organized interests and their public debate in the policy arena [3–7]. Hence, the main objective of this paper is to systematically record the diversity of stakeholder perspectives in Germany on synthetic fuels. The analysis is completed by means of a document-based positioning content analysis in order to represent the broadest possible spectrum of assessment of synthetic fuels. The leading research question, therefore, is: How do relevant stakeholder groups position themselves with regard to several aspects of synthetic fuels in Germany?

This paper summarizes the results of our analysis. First, Section 2 explains the method of the document analysis in more detail. Section 3 presents the main results of the content-related positioning analysis. Finally, Section 4 discusses the results and draws some conclusions.

## 2. Materials and Methods

The research applies a social-science-based document analysis by identifying and analyzing stakeholder documents found on the internet dealing with the topic of synthetic fuels. Desk research was completed by first identifying stakeholder documents published by stakeholder groups with reference to (synthetic) fuels. In the second step, we elaborated an assessment template ("stakeholder positioning profile") in order to carry out the empirical research. As a third step, the database was evaluated from a comparative perspective using the profile sheets filled out for every stakeholder.

Table 1 depicts a list of German stakeholders from the area of business, environment, and civil society. We found 17 stakeholder groups that published statements and positioning towards synthetic fuels.

**Table 1.** Compilation of the stakeholder organizations considered for document analysis.

| Economy |
|---|
| • German Association of the Automotive Industry (VDA) |
| • Association of the German Biofuel Industry (VDB) |
| • Federal Association eMobility \| Electromobility (BEM) |
| • Association of German Transport Companies (VDV) |
| • German Association of Energy and Water Industries (BDEW) |
| • Federal Association of Independent Petrol Stations and Independent German Mineral Oil Traders (bft/MEW) |
| • German Farmers' Association (DBV) |
| • Federation of German Industries (BDI) |

| Environment |
|---|
| • Federation for the Environment and Nature Conservation Germany (BUND) |
| • Germanwatch |
| • Greenpeace |
| • Nature And Biodiversity Conservation Union (NABU) |
| • Transport Clup Germany (VCD) |
| • World Wide Fund For Nature (WWF) |

| Society |
|---|
| • Agora Transport Transition (Agora) |
| • Federation of German Consumer Organizations (vzbv) |
| • German Automobile Club (ADAC) |
| • German Transport Club (VCD) |
| • Industrial Union of Metalworkers (IG Metall) |
| • Church Service in the World of Work (KDA) |

Source own elaboration.

Document identification focused on several types of stakeholder-related documents, which is position papers, statements, working reports, press releases, etc. Documents were

found issued by all 17 stakeholder groups. As a result, 42 documents were gathered for further content analysis. The processing of results was grouped by different sub-topics presented in the next section.

The data material from the documents was processed using a comparable analysis tool developed in advance, the "stakeholder positioning profile". The analysis tool contains a thematic structuring of relevant topics related to the positioning of fuels. First, it was formally ascertained whether associations position themselves explicitly or implicitly on fuels. Explicit positioning includes statements on fuels via documents such as statements, position papers, interviews, or press releases, while implicit positioning refers to other documents that do not contain an explicit statement of opinion (e.g., information material, issue papers, etc.). The content analysis first recorded the understanding of a transport turnaround from the association's perspective. Accordingly, the central goals, strategies, or characteristics of the associations' understanding of a transport turnaround were surveyed. The relevance of fuels was differentiated according to statements on biogenic and electricity-based or synthetic fuels. The topic area of aspects of synthetic fuels explored a large number of statements and positions through an in-depth analysis. These included assessments of the time dimension, areas of application, technology openness, production location, and economic viability, as well as the significance of the energy system, sustainability, and acceptance. The last main topic covers the positioning of the associations on political framework conditions.

## 3. Results

### 3.1. Overview of Positioning of Stakeholder Groups

The first result is a broad overview of several sub-topics among the 17 stakeholder groups. Table 2 indicates each organization with its statements concerning several aspects of synthetic fuels. It shows that stakeholder groups have a voice towards synthetic fuels and as such the topic is firmly anchored in German stakeholder communication. From that end, one may conclude that synthetic fuels—both biogenic and electricity-based fuels—are seen as an essential element for the mobility transition.

A deeper look reveals several sub-topics relevant to the field of synthetic fuels in the stakeholder debate. The large majority of stakeholders explicitly stated opinions and assessments on synthetic fuels within their documents. However, four associations did not position themselves explicitly—BDEW, VDV, VCD, KDA.

All but one association touched on the topic of the understanding of transport turnaround. Thus, synthetic fuels are an essential cornerstone in the context of the overall transition strategy of mobility. The relevance of synthetic fuels is addressed again by the large majority of stakeholders. Statements refer to both biogenic and synthetic fuels assessment and refer to different usage cases discussed. Different aspects of synthetic fuels are to a lesser degree addressed by the stakeholder groups. However, what is striking is the fact that different facets of synthetic fuels are touched, such as technology openness vs. selection, energy production and role of imports, assessment of economic efficiency, fuels for storage and flexibility, acceptance assessment, sustainability assessment, and the role of synthetic fuels in securing the economic future in the area.

**Table 2.** Overview of all stakeholders positioning according to the profile sheet.

| | Economy | | | | | | | | Environment | | | Civil Society | | | | | |
|---|---|---|---|---|---|---|---|---|---|---|---|---|---|---|---|---|---|
| | VDA | VDB | BEM | BDEW | VDV | bft/MEW | DBV | BDI | BUND | Greenpeace | WWF | Agora | vzbv | ADAC | VCD | IG Metall | KDA |
| **Topic: positioning on synthetic fuels** | | | | | | | | | | | | | | | | | |
| • Positioning: e = explicit, i = implicit | e | e | e | i | i | e | e | e | e | e | e | e | e | e | i | e | i |
| **Topic: mobility turnaround** | | | | | | | | | | | | | | | | | |
| • Understanding of mobility transition | ✔ | ✔ | ✔ | ✔ | ✔ | ✔ | | ✔ | ✔ | ✔ | ✔ | ✔ | ✔ | ✔ | ✔ | ✔ | ✔ |
| **Topic: relevance of synthetic fuels** | | | | | | | | | | | | | | | | | |
| • Biogenic fuels assessment | ✔ | ✔ | | | ✔ | | ✔ | ✔ | ✔ | | ✔ | ✔ | ✔ | ✔ | ✔ | | |
| • Synthetic fuel assessment | ✔ | ✔ | ✔ | ✔ | ✔ | ✔ | | ✔ | ✔ | ✔ | ✔ | ✔ | ✔ | ✔ | | ✔ | ✔ |
| • Areas of fuel usage | ✔ | ✔ | | ✔ | ✔ | | ✔ | ✔ | ✔ | ✔ | ✔ | ✔ | ✔ | ✔ | | ✔ | ✔ |
| **Topic: aspects of synthetic fuels** | | | | | | | | | | | | | | | | | |
| • Technology openness vs. selection | ✔ | ✔ | ✔ | ✔ | ✔ | ✔ | | ✔ | | | | ✔ | | | | ✔ | ✔ |
| • Energy production and role of imports | ✔ | ✔ | ✔ | ✔ | | ✔ | ✔ | ✔ | ✔ | | | ✔ | | | | | |
| • Assessment of economic efficiency | ✔ | | | ✔ | | ✔ | | ✔ | | ✔ | | ✔ | ✔ | ✔ | | | |
| • Fuels for storage and flexibility | ✔ | ✔ | | ✔ | | ✔ | | ✔ | ✔ | ✔ | ✔ | ✔ | ✔ | | | ✔ | |
| • Acceptance assessment | ✔ | ✔ | | | ✔ | | ✔ | ✔ | ✔ | | | ✔ | ✔ | ✔ | | | |
| • Sustainability assessment | ✔ | ✔ | ✔ | ✔ | | ✔ | ✔ | ✔ | ✔ | ✔ | ✔ | ✔ | ✔ | ✔ | | | |
| • Secure the economic future of the area | ✔ | | | | ✔ | | ✔ | | | | | | | | | ✔ | |

Source: own representation.

## 3.2. Understandings of a Transport Turnaround

Among the trade associations, different positions on the content of the transport turnaround can be identified. In a position paper supported by a total of eleven associations, the VDA and VDB advocate the transport turnaround in the sense of a sensible interplay between alternative drives and the internal combustion engine [8–10]. This shows a clear positioning in favor of the internal combustion engine, which is also seen as playing a key role in the transport transition in the future. The position is also supported by other associations—albeit not as explicitly—with an emphasis on the future of the combustion engine.

BDEW, BDI, and bft/MEW, for example, take a moderate position between alternative drives and the internal combustion engine. They argue that even a far-reaching transition to electromobility is not sufficient to achieve the climate protection targets and that alternative fuels are a necessary building block for climate protection [11–15]. The bft/MEW states: "Studies show that not only electromobility, but also efuels will represent a part of future mobility. This is the only way to achieve the German government's 2050 climate protection targets" [15]. For BDEW, climate protection in transport will only succeed "if alternative vehicle drives and fuels are used increasingly and consistently, with an 'energy turnaround in transport'" [16]. In contrast, the BEM positions itself, on the one hand, as envisaging a far-reaching technology shift from combustion engines to electric motors for the transport turnaround and, on the other hand, as seeing electromobility as a central building block for intermodal mobility chains [17].

Among the environmental associations, a different understanding of transport turnaround is evident. On the one hand, the feasibility of an ambitious, almost greenhouse gas-neutral mobility is brought to the fore (e.g., Greenpeace, WWF, BUND, NABU). Greenpeace goes furthest in this regard, considering emission-free mobility by 2035 to be feasible in a study commissioned by the Wuppertal Institute [18]. For Greenpeace, this requires a "completely different mobility", in which the sole conversion to electric cars is not a solution, but a clear hierarchy of measures along the focal points of avoidance, shifting and improvement is necessary [18]. BUND and WWF show a similar understanding. As an overall approach, sustainable mobility from BUND's perspective means "reducing the amount of traffic; it means honest prices, new mobility services, successful modal shift to rail and ship, but also fewer cars" [19]. However, both environmental organizations see a successful transport turnaround with a 95% $CO_2$ reduction only realized in 2050–and thus significantly less ambitious than Greenpeace.

Among the civil society associations, Agora understands the main task of a transport turnaround to be that of achieving the goals of the federal government. Agora cites two pillars in particular on which the transport turnaround rests: (1) the mobility turnaround and (2) the energy turnaround in transport [20–22]. The vzbv primarily focuses on the consumer perspective, as for "consumers, sustainable and affordable mobility is the basis for welfare, quality of life and social participation" [23]. The vzbv, therefore, advocates measures with low costs and comprehensive benefits for consumers in the transport sector [23]. The ADAC supports the transport turnaround and sees the following goals in the foreground: securing individual mobility, limiting greenhouse gas emissions, reducing dependence on fossil resources, and reducing air pollutants [24]. In order to achieve these goals of the transport turnaround, a holistic approach is necessary that focuses on alternative drive systems, fuels based on renewable energies, the optimization and emission reduction of conventional drive systems, the integration and networking of transport modes, and modern mobility services. IG Metall's approach combines climate neutrality with consumer and worker interests [25–27]. In the adopted action program, four central fields of action with formulated core demands are identified for the next four years [25]. These are the ramp-up of electromobility, entry into the mobility turnaround, determined advancement of the energy turnaround, and new concepts and instruments for regional and structural policy.

### 3.3. Relevance of the Alternative Fuels

Passenger and freight transport powered by fossil fuels contributes significantly to climate change through the emission of greenhouse gases. However, the predominantly used petrol and diesel fuels can also be produced regeneratively–as so-called "synthetic fuels"–from non-fossil carbon sources such as biogenic residues in combination with the direct conversion of $CO_2$ and renewable hydrogen and can thus help to reduce $CO_2$ emissions. With regard to the future role of the different types of fuel, the stakeholders have different assessments and positions.

### 3.3.1. Biofuels

In the view of most trade associations, including VDA, VDB, DBV, the use of advanced biofuels (second generation) from waste wood, straw, waste, and residual materials is becoming increasingly important for the future defossilization of the transport sector [1,9,10]. From the DBV's point of view, biofuels that provide proteins and other high-value feedstuffs in co-production should be preferred [28]. However, the DBV criticizes the greenhouse gas accounting according to the source principle because the emissions from raw material production are accounted for in agriculture while the greenhouse gas avoidance is attributed to the transport sector. These should be attributed at least proportionately to agriculture. The BDI, on the other hand, is of the opinion that the production of biofuels from energy crops should be reduced on the basis of sustainability criteria [12]. Even though it is recognized that biofuels can be used very effectively for climate protection, numerous environmental and civil society associations express concerns with regard to environmental problems—especially for first-generation biofuels [29–31].

Some environmental organizations even call for the phase-out of fuel production based on cultivated biomass by 2030 [31,32]. Instead, the limited amount of biomass from residues should be used in other sectors outside transport in a much more climate-friendly way. Other institutions, such as Agora, doubt that biofuels alone have the potential to cover the future energy demand in transport [20]. From the vzbv's point of view, biofuels only make sense if they avoid side effects such as deforestation of rainforests or food competition and the use of pesticides and fertilizers. In its comments, the vzbv refers in particular to first-generation biofuels and does not address advanced biofuels (from the second generation onwards). The ADAC and the VCD see future potential for second-generation biofuels and the ADAC also temporarily for CNG [33,34]. In its information document "Designer fuels", the VCD weighs up the advantages and disadvantages of BtL (biomass to liquid) as a second-generation biofuel [35], which are explained in more detail in the following section.

Thus, in the view of most associations, the use of advanced biofuels (second generation) from waste wood, straw, waste, and residual materials can play an important role in the future defossilization of the transport sector. Numerous environmental and civil society associations express concerns about environmental issues such as rainforest deforestation or food competition and the use of pesticides and fertilizers.

### 3.3.2. Electricity-Based Fuels

In the stakeholder documents, electricity-based fuels are detailed as hydrogen production via electrolysis of water with the help of–ideally–renewable (surplus) electricity. In a further step, hydrogen is combined with carbon dioxide from industrial processes or the ambient air. This produces electricity-based fuels such as synthetic diesel, synthetic petrol, and synthetic gas. While the environmental associations mostly outline very explicitly possible production routes for electricity-based fuels in their documents, electricity-based fuels are taken up in the documents of the civil society associations from hydrogen and $CO_2$, but without outlining details explicitly.

With regard to the future role of electricity-based fuels in the transport sector, most civil society associations consider them important, stressing that they should be used primarily in areas where direct electrification seems difficult [26]. This applies, for example, to aviation and shipping, where the combustion-based drive train continues to be technically realized and maintained for lack of alternatives. Its use as a raw material for the chemical industry is also envisaged [25]. However, most associations highlight a comparatively inefficient use of electricity by means of electricity-based fuels, which amounts to a five- to six-fold higher energy demand compared to a battery-electric car using electricity directly [23]. From IG Metall's point of view, electric cars and hydrogen cars will diffuse into the road transport market for the time being due to the additional work step for the production of the synthetic fuel as well as its energy consumption [27]. The ADAC as well as the vzbv are rather skeptical about an early market introduction of electricity-based fuels, as they are currently still at the research stage (e.g., PtX). The

disadvantages cited are a lack of efficiency (in "well-to-wheel" terms, approx. 10–15% compared to 70–80% for electric cars) and high costs, currently approx. 4.50 euros for production [36]. IG Metall also takes a critical view of an early market launch. In terms of an industrial policy in which Germany remains a development and production location in the future, IG Metall believes that industry needs "technologically achievable capacities for battery cells or synthetic fuels, but has so far not managed to organize the necessary investments and upfront investments", because the "development and industrialization of alternative synthetic fuels remains on the agenda, even if only for aircraft, ships or as a raw material for the chemical industry" [26].

Some civil society associations also point out that it must be guaranteed that renewable electricity is generated from renewable energy sources so that the fuels contribute to decarbonization [20]. Reference is therefore made to $CO_2$-neutral use, because "combustion in the car then produces $CO_2$, but only as much as was previously taken from the air for the production of the e-fuels" [27].

This view is also shared by environmental organizations (e.g., BUND, Greenpeace, WWF). They also recognize that in order to achieve the goal of a complete reduction of greenhouse gas emissions in the transport sector, gas, and liquid fuels on a renewable basis must be used in addition to electricity from renewable energies, e.g., for trucks, long-distance buses, ships, in air traffic or also to cover the remaining liquid fuel demand of passenger cars and smaller trucks with plug-in hybrid or range extender drives. Even with ambitious measures to be implemented as a priority to shift, avoid, and increase efficiency in the transport sector, there will still be a "residual demand" for liquid or gaseous fuels in the transport sector in 2050 [31]. From the point of view of the environmental associations, however, PtX products should only be used where no direct electricity option is available. This is given in usage areas of basic material in the chemical industry, storage to ensure the stability of the energy supply and special transport areas such as shipping and aviation.

PtX production does not rely on the provision of $CO_2$ from fossil sources. In principle, in the view of the association, the extraction of the $CO_2$ required for the production of synthetic fuels should take place from the air. The use of PtX substances in the context of universal fuel blending is excluded due to the use-specific efficiency test. This is because, according to BUND, the highest premise for the success of the energy transition—to drastically reduce energy consumption—must remain in place [30]. Greenpeace also emphasizes that electricity-based fuels are significantly more expensive and less efficient in the long term than running cars directly on electricity.

Most industry associations regard electricity-based fuels as an important building block for the defossilization of the transport sector. In the view of the VDA, the development of capacities for the production of electricity-based fuels is already necessary today in order to achieve the EU climate protection target 2030 for the transport sector (minus 30% compared to 2005) [37]. The VDA sees an important contribution from synthetic fuels in particular. The BDEW implicitly positions itself on electricity-based fuels, seeing the future of climate-friendly transport primarily in the use of renewable energies, i.e., directly or indirectly on the basis of renewable electricity, e.g., hydrogen or other PtX as well as biogas [16]. Unsurprisingly, the BEM is the only trade association to take a different position with regard to electricity-based fuels, emphasizing that these are produced by an energy-intensive process that is not feasible today on the basis of renewable energies for the requirements of individual transport and that requires large amounts of energy [17]. MEW/bft see electricity-based fuels as a "bridge" and recur in their position to the aspects of acceptance and infrastructure: "It may be that the efficiency from generation to consumption is somewhat worse when using eFuels than the use of pure electric drives. But the decisive factor will be consumer acceptance. And this is certainly incomparably greater for a gentle migration process via eFuels, at the end of which, in my opinion, there must be the use of hydrogen anyway" [13]. In the view of the BDI, electricity-based fuels make an immediate contribution to defossilization in all transport sectors and are necessary for achieving the Paris climate targets [12]. In this context, they should complement

electrification or electromobility by being used in particular in transport sectors where electrification is not possible, such as air or sea transport. The BDI also emphasizes some disadvantages associated with electricity-based fuels, such as the increased energy demand that can be met by importing from regions with a surplus of renewable energy.

Summarizing the individual positions, different positions can be deduced with regard to electricity-based fuels. In the long run, electricity-based fuels could play a limited but useful role in the overall system in 2050 to cover the remaining final energy demand of the transport sector. The majority of transport applications for electricity-based fuels are seen in trucks, long-distance buses, ships, and air traffic. The use of electricity-based fuels in the passenger car sector is only addressed by very few actors in the existing or new vehicle sectors and remains controversial.

### 3.4. Aspects of Electricity-Based Fuels

3.4.1. Location of Energy Production and Relevance of Imports

Site selection and location of production plants are key issues in the area of synthetic fuels. A key variable for economic efficiency is peak load hours per year for renewable energy production. For that reason, most stakeholders favor abroad production capacities and corresponding imports of synthetic fuels.

Trade associations such as the VDA state that renewable electricity volumes can be produced in Europe [1]. A major advantage of European production relates to the existing infrastructure which consists of a widespread grid of public filling stations, and the distribution systems of fuel retailers could continue to be used [10]. In addition, according to the BDEW, there is room for flexibility by coupling renewable energy use with fuel production and thus making grid use more efficient [11,16].

The MEW states that the energy industry needs to be placed in a market economy [13]. On the other side of the spectrum, BEM regards synthetic fuels as a pretext to continue selling conventional passenger cars, regardless of energy production: "with the help of the permit, it is possible for conventional car manufacturers to keep their old products on offer and sell the engines with efuel admixtures as environmentally friendly" [17]. This is because by blending with synthetic, combustion engines could be sold as "environmentally friendly" even though they are less efficient than battery-powered drives and also perform poorly in the energy balance [17].

The environmental stakeholder groups have some clear positions on the location of synthetic fuel production. The BUND states that water supply in warm but water-scarce areas needs to be assessed and sustainable land use should be ensured, in case synthetic fuels or renewable energy imports are foreseen [32]. Agora [20] argues that the electricity needed for synthetic fuel production relies heavily on imports. A study issued by Agora Energiewende [21] develops several scenarios with production sites in North Africa, the Middle East, Iceland (geothermal, hydropower), and Germany (wind offshore North Sea and Baltic Sea) with a view to costs. However, no clear position is taken with regard to the best site location policy.

3.4.2. Evaluation of Economic Efficiency

The economics of synthetic fuels addresses in particular the current and future production costs. The VDA and the BDI, for instance, question the economic viability. They criticize the lack of openness to technological ideas that prevents necessary investment in synthetic fuel technology development [1,12]. The VDA explicitly calls for investment security [1]. The MEW takes a different view, describing synthetic fuels as a win-win situation from an economic perspective [15]. To their opinion, synthetic fuels have several advantages such as climate neutrality, storability, and use in conventional engines. With regard to manufacturer costs, the VDA foresees a competitive cost level of ca. 1 euro per liter of diesel equivalent (for the time being ca. 4.50 euros per liter) [1] From the MEW's and BDI's point of view, synthetic fuels could be manufactured at a price level between 0.70 euros and 1.30 euros per liter in the year 2050 [12,15].

Greenpeace is the only environmental organization to comment on the economic viability of the price development of synthetic fuels: "E-Fuels will be significantly more expensive and less efficient in the long term" [18]. The environmental stakeholder groups (i.e., WWF and BUND) did not make a point on economic viability. Civil society associations, such as the vzbv and the ADAC, were mostly critical concerning economic viability [23,24,36]. They argued that current transportation policy favors heavily electric cars to the disadvantage of synthetic fuels [23]. However, it was clearly stated that electric cars and electric drives have considerable advantages in terms of efficiency advantages and cost-effectiveness [19,23]. Considering the future costs of synthetic fuels, some consider a price of 2.29 euros including taxes as reasonable [36]. A detailed cost analysis is provided by the study of Agora [21]. They analyze the cost structure of synthetic fuels by means of scenario research. Depending on the time periods considered (2020, 2030, 2050) and the location of energy or fuel production (North Africa, Middle East, Iceland (geothermal, hydropower), Germany (wind offshore North Sea and Baltic Sea), cost estimations may vary considerably between 10 and up to more than 25 cents/kWh for 2020, between about 10 and 20 cents/kWh for 2030 and between ca. 8 and 15 cents/kWh for 2050.

### 3.4.3. Synthetic Fuels and Energy Storage and System Flexibility

Several stakeholder groups address the role of synthetic fuels for energy storage and energy system flexibility. Trade associations such as VDA and VDB argue in favor to use synthetic fuels as a chemical storage option for surplus electricity [1,9]. According to the VDB, biorefineries are very much suitable as storage providers and serve as inexpensive "long-term batteries" that will help to stabilize the power grid. The advantage to use synthetic fuels for grid stabilization is also held by the BDEW, as PtX can additionally contribute to flexibilization and security of supply in the energy sector [6]. The BDI advocates for long-term storage of renewable energy in the natural gas grid or in liquid fuel storage facilities.

Environmental groups such as WWF, BUND, and Greenpeace recognize the storage option of using electricity surpluses within synthetic fuel production. However, WWF details the fact that electricity surpluses occur intermittently while, on the other side, electrolyzers need to constantly operate on a full-time level for continuous production [31]. Civil society groups such as vzbv recognize synthetic fuels as a "relatively new energy storage system" while the union IG Metall speaks on a very general level of "a storage technology" [23,26]. Agora mentions the issue of energy storage by PtX technologies, while they observe considerable efficiency deficits.

### 3.4.4. Acceptance Assessment

Social acceptance is a major issue for synthetic fuels since they need consumer demand at the filling stations. Due to the relatively small structural changes that synthetic fuels entail, it is expected that these fuels will meet with high acceptance among consumers [15]. A key advantage is the existing filling station infrastructure [1,10]. The BDI is resistant to specific acceptance statements on synthetic fuels, although it does state that synthetic fuels contribute to "increasing flexibility in the ramp-up of electrification and possible hedging through diversification when risks arise with regard to the expansion of the electricity grid/charging infrastructure, resource availability, customer acceptance, battery price development or recycling" [12].

Environmental stakeholder groups in general have little to say on social acceptance. Just one statement is found by the BUND. For BUND, PtX materials are acceptable only if renewable energy is used in their production in terms of climate policy [30]. Greenpeace sees at least partial acceptance among consumers. They cite a survey conducted by YouGov on behalf of the Deutsche Presse-Agentur. According to the survey, more than half of the respondents judge synthetic fuels as an alternative to electric cars [38] in case the price is competitive: "For just under half of the respondents, so-called e-fuels produced with green electricity should cost less than 1.50 euros per liter. 28 percent would find 1.5 to

2 euros okay. This means that e-fuels should generally not cost much more than a liter of premium petrol, the price of which was 1.53 euros in May according to the Mineral Oil Association" [38]. A similar positioning is undertaken by ADAC. In the association's view, most consumers will accept synthetic fuels, given that the end-consumer price level is affordable, nevertheless, there is high environmental awareness among consumers [24]. In its documents, the vzbv calls for "political and technical options to be carefully weighed up and unacceptable side-effects to be ruled out before proceeding to implementation. Otherwise, there will be no acceptance for biofuels, also for ecological reasons [29]".

### 3.4.5. Role of Securing Germany as a Location for Business and Investment

Securing future economic competitiveness is a major objective of national economic and innovation policy. Thus, it is not surprising to see positioning statements concerning synthetic fuels and the role of securing Germany's economic profile. Economic stakeholder groups outline the importance of establishing parts of domestic value chains in Germany for synthetic fuels. MEW, for instance, sees potential for at least part of the value creation in Germany as secured by means of synthetic fuels [13]. The BDI calls for safeguarding knowledge and jobs "in core European industrial technologies such as engines and gear-boxes as well as in the energy and gas industry" [12] while the trade union IG Metall lays emphasis to guarantee Germany as a key innovation and production place for synthetic fuels and battery cells for electric drives [26]. All in all, stakeholder positioning concerning Germany's role as a production site for synthetic fuels was rare; environmental and civil society groups did not focus on these issues.

### 3.4.6. Assessment of Sustainability Criteria

Sustainability assessment of the synthetic fuels value chain, in contrast, has been very popular in stakeholder positioning. In fact, it has been touched by all stakeholder organizations except four. Since sustainability is conceptually based on the three main fundamental topics of ecology, economy, and social issues, one may expect that many stakeholders address the issue–albeit with varying emphasis. Trade associations highlight the fact that synthetic fuels contribute to greenhouse gas reduction and thus play an important role to meet the Paris climate goals [9,12,13,17]. Both the VDA and the DBV request compliance with sustainability for biogenic fuels so that there are "no undesirable environmental impacts" [39]. The DBV criticizes that greenhouse gas accounting according to the so-called source accounting is inappropriate since emissions from raw material production are allocated in the area of agriculture while greenhouse gas avoidance is assigned to the mobility sector.

Environmental stakeholder groups call for far-reaching consideration of sustainability criteria along the whole value chain of synthetic fuels. BUND points out that the sustainability of water needs to be considered due to the fact that locations with high levels of renewable energy production via wind and solar often face water scarcity. In addition, the association calls for social-ecological land use to be taken into account in the context of fuel production and for further work on sustainability criteria for PtX in the future [19,32,40]. Greenpeace [41] also emphasizes from a climate protection perspective that "synthetic (liquid) fuels [have] a worse efficiency than propulsion systems with hydrogen fuel cells. In this respect, it would be very welcome from a climate protection perspective that vehicle concepts with fuel cells for large cars and trucks or long distances become marketable as soon as possible." The vzbv emphasizes that full-scale renewable electricity production is essential for the sustainability of synthetic fuels. Agora Verkehrswende also highlights the need for renewable electricity as an important condition to contribute to decarbonization in electricity-based fuel production [42,43]. The ADAC, on the other side, addresses demand-side usage patterns noting that synthetic fuels should be used in the existing fleet of cars and trucks since they expect today's vehicles will remain in the fleet for a considerable time span [36].

## 4. Discussion and Conclusions

The document-based research of the German stakeholder discourse on synthetic fuel resulted in a positioning and narrative analysis. What is striking, is the fact that communication on synthetic fuels is deeply rooted in the stakeholder discourse. All identified stakeholder organizations have expressed opinions and positions toward synthetic fuels with a great scope of different topics, explanations, and evaluations. What can be seen are quite different narratives and positioning patterns across the sample. These different assessments cover both the big picture with different approaches on how to implement and manage the overall mobility transition towards climate friendliness, and detailed evaluation on a variety of aspects of synthetic fuels such as single issues of technical, economic, social, and ecological aspects of the value chain.

Taking the disparity of single results altogether, three cross-stakeholder narratives can be synthesized. The main frames and narratives comprise individual aspects mentioned and condense them in a sharpened narrative. These three narratives serve as distinctive features for illustrating the patterns of stakeholder debate and discourse in Germany.

**Narrative 1: Synthetic fuels as a key component for the mobility transition**

Within the first frame, synthetic fuels serve as a key component for the success of the mobility turnaround. In that sense, they contribute essentially to reaching climate neutrality via a substantial reduction of greenhouse gases. Used for the existing fleet but also beyond, synthetic fuels are a relatively easily available solution to be implemented in existing value chains and their infrastructure. From that end, synthetic fuels are an important building block for achieving climate targets. By dovetailing the two sectors of energy and transport, some further energy system advantages become obvious: synthetic fuels may contribute to grid stability, system flexibility, and security of energy supply. Synthetic fuels, when produced with renewable surplus electricity, may store energy in a sense of a grid-stabilizing and fluctuation-eliminating application. In addition, national economic and industrial policy goals can be reached by locating at least part of the value chain domestically which serves social and economic benefits along the transformation process. The added value of domestic value chains relates to local value creation, know-how and job security, purchasing power, and tax revenues. The acceptance of end-users is a crucial success factor for synthetic fuels within the existing car fleet and beyond. The possibility of using the existing infrastructure, which does not require any significant changes in consumption patterns will favor market penetration of synthetic fuels. A key usage advantage is seen in areas of application where no other technical solutions are available for the moment such as in ship and aircraft traffic, but also in vehicle traffic for private transport in rural areas and for use in the existing car and truck fleet.

If we break down this narrative into its individual positionings elements, the following argumentation is central:

- Synthetic fuels as an immediately available building block for the defossilization of the transport sector
- Synthetic fuels as stabilizers and storage for the energy transition
- Synthetic fuels for securing locations in the national transformation process
- High relevance of openness to technology
- End-user acceptance is a crucial element and can be expected
- Synthetic fuels relevant for motorized private transport in particular in rural areas

**Narrative 2: Synthetic fuels as an essential strategic niche management component**

The second frame foresees synthetic fuels as an essential niche management component. In this understanding, the overall strategy for the mobility transition is on alternative drives wherever this is possible and reasonable. The change towards electric drives for a wide range of modes of transport is the key element while synthetic fuels serve for specific niches. As a niche element, synthetic fuels will only be used where no direct electric drive option is possible due to technical efficiency reasons. Thus, usage patterns refer to the user as a basic material in the chemical industry, as an energy storage option serving system stabilizing and flexibility reasons, or as an energy source in special transport modes such as

shipping, aviation, and special vehicles (e.g., agriculture, road and building construction, etc.). In this frame, niche application usage refers to advanced synthetic fuels based on electric-based fuels while biomass-based fuels shall be phased out by the year 2030. Several requirements for synthetic fuel production are being imposed: the use of $CO_2$ from the air capture for climate policy reasons (no fossil $CO_2$); the use of 100% renewable energy for fuel production; the long-term phase-out of biomass-based fuels while in short-term only second-generation biofuels with no to little ecological impact are acceptable. In case biofuels are used, limited to no use of pesticides, fertilizers and the endangering of the food industry as well as rainforest deforestation must be avoided. Further sustainability requirements refer to water usage and land use for renewable energy production. On a timeline, synthetic fuels are foreseen after 2030 at the earliest. The next few years should be used for further research and development in the field in order to reach advanced technology readiness levels with high levels of efficiency.

The individual arguments of this narrative can be summarized as follows:

- Potential of synthetic fuels exists in principle
- Areas of application only in those areas where electrification is difficult to implement
- $CO_2$ sourced from the air for the production of synthetic fuels
- Renewable energies as the only basis for energy production
- Establish and focus on sustainability rules
- Only use synthetic fuels in practice after 2030

**Narrative 3: Mobility transition as sustainable, affordable, safe, and comfortable mobility– with or without synthetic fuels**

The third frame poses different requirements for a successful mobility transition and defines subsequently from these claims the role of synthetic fuels. The main emphasis here is on social aspects for citizens and consumers as essential target groups in the mobility transition. A successful transition needs to be based on sustainable, comfortable, and affordable mobility that is accessible to all people as the basis for welfare, quality of life, and social participation and involvement. Benefits for citizens also include better air and higher quality of life in densely populated areas, less congestion on the roads, and more space for urban culture, new jobs, etc. However, there is a trade-off as energy and resource consumption, and greenhouse gas emissions must become significantly more expensive, while at the same time ensuring a social balance. While sustainable mobility will most probably become more expensive, it needs to be assured that the high level of mobility for all citizens can be maintained. Encouraging modal shift and decongesting transport corridors needs to rely on the innovation of intermodal and interoperable transport.

Thus, synthetic fuels do play a subordinate role in the mobility transition. If they serve sustainable, affordable, safe, and comfortable mobility, they are welcome. Affordability, for instance, relates to prices. Most users will only accept synthetic fuels at an adequate price level at filling stations. Since production costs will expectably be high, tax and due policy will be essential in order to guarantee affordable price levels. On the supply side, there is a need to ensure openness to technology with policies available to encourage framework conditions for investment strategies that stimulate investors and companies to invest in the development and production of synthetic fuels.

This narrative can be broken down into the following argumentations:

- Sustainable, affordable, safe, and comfortable mobility is the basis for welfare, quality of life, and social participation
- Use of synthetic fuels to reduce $CO_2$ emissions, especially where battery-electric mobility is not an option
- Set price incentives and at the same time mitigate undesirable side effects and ensure mobility as social participation
- Intensify research and development on synthetic fuels and focus on economic efficiency
- Ensure acceptance with regard to the users of the fuels
- Involve investors, businesses, and employees in regional development plans.

**Author Contributions:** Conceptualization, D.S. and L.S.; methodology, D.S. and L.S.; investigation, D.S. and L.S.; writing, D.S.; review and editing, D.S.; supervision and project administration D.S. All authors have read and agreed to the published version of the manuscript.

**Funding:** This research was funded by the "Ministerium für Wissenschaft, Forschung und Kunst Baden Württemberg". We also thank the "Ministerium für Verkehr Baden-Württemberg" and the "Strategiedialog Automobilwirtschaft BW" for sponsoring the project "reFuels—rethinking Fuels".

**Data Availability Statement:** The data presented in this study, if not included in this paper, are available on request from the corresponding author.

**Conflicts of Interest:** The authors declare no conflict of interest.

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
