# Peer review of "Stakeholder Discourse on Synthetic Fuels: A Positioning and Narrative Analysis"

_2673-3994, doi:10.3390/fuels4030017_

Round 1
Reviewer 1 Report
The article is interesting and important due to the subject of alternative fuels, including hydrogen. Various groups of fuels have been discussed, and the following concepts appear in the article: alternative fuels, synthetic fuels, biofuels, e-fuels.: alternative fuels, synthetic fuels, biofuels, e-fuels. Recently, these terms have been used in numerous publications, however, their meaning is understood differently. In my opinion for the clarity of the article, it is advisable for the authors to define these terms in the introduction.

Author Response
Dear Reviewer,
many thanks for reviewing our article. We are happy that your overall assessment is positive. And we are grateful for your advice to include definitions on fuel terms. We included a definition paragraph in the introduction sector.
Best wishes
Reviewer 2 Report
The paper is extremely interesting because is a correct analysis, based on documents, on the opinion of important stakaholders on the future of liquid fuels in the context of energy transition. The paper is well discussed and has a large interest for researchers and not only. The literature is properly cited. A very interestting paper which can help the discussion on the field. The paper can be accepted in the present form.
Author Response
Dear Reviewer.
many thanks for reviewing our paper. We are very happy for your overall very positive assessment and your conclusion that the paper can be published in the present form.
Best wishes
Reviewer 3 Report
Dear Authors,
After carefully evaluating the content, the paper could benefit from a more scientific approach and a more focused scope that aligns with the expectations of MDPI Fuels publication.
Additionally, the introduction could benefit from adding references and a general improvement in the text.
Although the analysis of the diversity of stakeholder perspectives on synthetic fuels and their discourse is interesting, it falls outside the primary scope of the publication.
I appreciate the effort put into analyzing 41 sources from 17 stakeholders from the economy, environment, and civil society and synthesizing the three narratives dominating the German discourse on synthetic fuels.
However, to make the article more suitable for publication in MDPI Fuels, I suggest focusing on a more scientific and specific aspect of synthetic fuel research. I would be happy to provide further feedback and suggestions for revisions to improve the paper for consideration in this publication.
Author Response
Dear reviewer,
many thanks for your thourough and deep review of our paper and your helpful hints to improve the paper. We follow your recommendations and reworked the paper with the following:
- we inserted in the introduction a paragraph with definitions on several fuels terms,
- we inserted in the introduction a linkage from our approach to teh field of risk and technology reserach and acceptance research including references
- in the methods section we inserted a paragraph on how we analysed more in detail the documents with the "Association Position Profile".
We hope this improves the paper and is in line with your recommendations.
Best wishes
Round 2
Reviewer 3 Report
Dear Authors,
Thank you for considering my suggestion and accepting the article in its present form. Your article provides valuable insights on the topic and will be a valuable addition to the academic discourse.
I look forward to seeing the impact of your work in the field.
Best regards,